# Current Status and Future Prospects of Stem Cell Therapy for Infertile Patients with Premature Ovarian Insufficiency

**DOI:** 10.3390/biom14020242

**Published:** 2024-02-19

**Authors:** Hye Kyeong Kim, Tae Jin Kim

**Affiliations:** 1Department of Obstetrics & Gynecology, Infertility Center, CHA University Ilsan Medical Center, Goyang 10414, Republic of Korea; hazlendhye1@chamc.co.kr; 2Department of Urology, CHA University Ilsan Medical Center, CHA University School of Medicine, Goyang 10414, Republic of Korea

**Keywords:** premature ovarian insufficiency, premature ovarian failure, infertility, stem cell therapy, regenerative medicine

## Abstract

Premature ovarian insufficiency (POI), also known as premature menopause or premature ovarian failure, signifies the partial or complete loss of ovarian endocrine function and fertility before 40 years of age. This condition affects approximately 1% of women of childbearing age. Although 5–10% of patients may conceive naturally, conventional infertility treatments, including assisted reproductive technology, often prove ineffective for the majority. For infertile patients with POI, oocyte donation or adoption exist, although a prevalent desire persists among them to have biological children. Stem cells, which are characterized by their undifferentiated nature, self-renewal capability, and potential to differentiate into various cell types, have emerged as promising avenues for treating POI. Stem cell therapy can potentially reverse the diminished ovarian endocrine function and restore fertility. Beyond direct POI therapy, stem cells show promise in supplementary applications such as ovarian tissue cryopreservation and tissue engineering. However, technological and ethical challenges hinder the widespread clinical application of stem cells. This review examines the current landscape of stem cell therapy for POI, underscoring the importance of comprehensive assessments that acknowledge the diversity of cell types and functions. Additionally, this review scrutinizes the limitations and prospects associated with the clinical implementation of stem cell treatments for POI.

## 1. Introduction

Premature ovarian insufficiency (POI), also known as primary ovarian insufficiency, premature ovarian failure (POF), or premature menopause, denotes the partial or total loss of normal ovarian function before 40 years of age. While “premature ovarian failure” and “premature menopause” have historically been used interchangeably with “premature ovarian insufficiency”, the latter term is now more favored due to the potential persistence of impaired ovarian function over varying durations. Moreover, “failure” or “menopause” can convey negative connotations. Additionally, “primary ovarian insufficiency” is synonymous with “premature ovarian insufficiency” [1,2].

The triad that defines POI encompasses oligomenorrhea or amenorrhea, heightened serum gonadotropin levels, and diminished serum estrogen levels. Conventionally, POI is diagnosed when oligomenorrhea or amenorrhea persists for more than 4 months and the serum levels of follicle-stimulating hormone (FSH) on at least two consecutive tests exceed 25 mIU/mL at 4-week intervals in women aged under 40 years [2,3]. Estrogen deficiency preceding natural menopause causes distressing symptoms, such as hot flashes, night sweats, sleep disruptions, vaginal dryness, anxiety, and depression. Furthermore, it is correlated with an increased risk of osteoporosis, cardiovascular disease (CVD), and to some extent, neurodegenerative aging [4].

Infertility is commonly accompanied by POI, which results from premature follicle destruction and early depletion of ovarian follicles. Approximately 1% of women of childbearing age are affected [5], with only 5–10% conceiving naturally and delivering a child after POI diagnosis [6]. Presently, oocyte donation or adoption is deemed the most effective option; however, individuals with POI strongly desire genetic offspring. Traditional infertility treatments have shown limited efficacy, with ovulation induction yielding an overall pregnancy rate of 6.3%, whereas controlled studies on gonadotropin-releasing hormone agonist (GnRH-a) suppression versus placebo revealed no statistically significant difference in pregnancy rates [7,8,9,10].

Recently, stem cell (SC) therapy, which has shown tremendous success in cord injury, neurodegenerative diseases, multiple sclerosis, heart diseases, and diabetes mellitus, has been promising for restoring the diminished ovarian reserve in patients with POI [11,12,13]. The regenerative potential of SCs, attributed to their self-renewal ability and pluripotency, makes them a compelling therapeutic option. This review provides a succinct overview of current methods of POI diagnosis, evaluation, and treatment, shedding light on innovative stem cell therapy approaches and addressing the associated challenges and limitations.

## 2. Etiology and Causes of Premature Ovarian Insufficiency

### 2.1. Chromosomal and Genetic Factors in the Clinical Setting of POI

Traditionally, more than 70% of POI cases were considered idiopathic [2,6]. However, recent advances in whole-genome sequencing have revealed that at least 30% of these cases originate from genetic causes [4]. Therefore, after carefully evaluating the causative factors, idiopathic POI can be diagnosed by exclusion. 

#### 2.1.1. Sex Chromosome Abnormalities

Previous research has indicated that approximately 10–12% of patients with POI exhibit chromosomal abnormalities, the majority of which involve the X chromosome [2,14,15], with only a small fraction associated with autosomal chromosome defects [7].

Turner’s syndrome (TS), characterized by a 45, XO karyotype, is the most prevalent, occurring in 1:2500 live births. TS results from the partial or complete depletion of one X chromosome through deletion, translocation, inversion, isochromosome formation, or mosaicism [16]. In cases of mosaic TS (45, X/46, XX), individuals often display milder clinical symptoms, with the severity being directly linked to the extent of the mosaicism. Notably, approximately 12% of patients with TS have a Y chromosome, which is sometimes accompanied by an SRY gene mutation. Given the strong association between the Y chromosome in women with TS and gonadal malignancy, gonadectomy is recommended upon detecting the Y chromosome [2,7]. 

In addition to TS, other X chromosomal abnormalities in patients with POI encompass duplications, deletions, and translocations of the X chromosome [17]. The notable frequency of chromosomal abnormalities in patients with POI warrants chromosomal analysis to be conducted for all patients with non-iatrogenic POI, irrespective of age [2,14]. 

Fragile X syndrome, an X-linked dominant disorder, is caused by a mutation in the *fragile X mental retardation 1* (*FMR1*) gene, located on the X chromosome. The *FMR1* gene encodes a polymorphic CGG trinucleotide repeat in its 5′ untranslated region. A mutation in *FMR1*, specifically an expansion of the CGG repeats exceeding 200, leads to silencing of the *FMR* gene, resulting in deficient production of the FMRP protein. When the number of increased repeats falls between 55 and 200, it is termed a premutation of fragile X. Females carrying the fragile X premutation have a notable association with POI [18], ranging from 13 to 26% [19]. The prevalence of POI among women with sporadic fragile X premutations varies from 0.8 to 7.5%. However, the incidence rises to 13% in women with fragile X premutation who have a family history of POI [20,21]. Given the substantial prevalence of fragile X premutations and their consequential impact, its testing is necessary for all women diagnosed with POI. 

Additionally, various candidate genes on the X chromosome, including *ubiquitin-specific peptidase 9 X-linked (USP9X)*, *zinc finger protein X-linked (ZFX)*, *bone morphogenetic protein 15 (BMP15)*, and *short-stature homeobox gene (SHOX)*, have been elucidated [22]. Among those genes, *BMP15*, *progesterone receptor membrane component 1 (PGRMC1)* and *FMR1* are correlated with non-syndromic POI [23]. However, their specific roles in the development of POI remain unestablished.

#### 2.1.2. Autosomal Chromosome Abnormalities

The progress of genetic analysis technology has identified many autosomal genes as factors that are implicated in POI. Mutations participating in POI were identified for some of these genes, whereas others were considered candidate genes. Currently, researchers have primarily investigated genes associated with various aspects of ovarian development and function. These encompass gonadogenesis, featuring the involvement of *leucine-rich repeat-containing G protein-coupled receptor 4 (LGR4)* and *PR domain-containing 1 (PRDM1)*. In meiosis, the pivotal genes include *cytoplasmic polyadenylation element-binding protein 1 (CPEB1)*, *KASH domain-containing 5 (KASH5)*, *minichromosome maintenance domain-containing 2 (MCMDC2)*, *meiosis initiator (MEIOSIN)*, *nucleoporin 43 (NUP43)*, *ring finger and WD repeat domain 3 (RFWD3)*, *shortage in chiasmata 1 (SHOC1)*, *SLX4 structure-specific endonuclease subunit (SLX4)*, and *stimulated by retinoic acid gene 8 (STRA8)*. Moreover, autosomal chromosomal genes, such as *arachidonate 12-lipoxygenase, 12S type (ALOX12)*, *bone morphogenetic protein 6 (BMP6)*, *H1.8 linker histone (H1-8)*, *hyaluronan-mediated motility receptor (HMMR)*, *hydroxysteroid 17-beta dehydrogenase 1 (HSD17B1)*, *macrophage-stimulating 1 receptor (MST1R)*, *protein phosphatase 1B*, *magnesium-dependent, beta isoform (PPM1B)*, *zygote arrest 1 (ZAR1)*, and *zona pellucida glycoprotein 3 (ZP3)*, play roles in folliculogenesis and ovulation. Additionally, investigations into genes involved in steroidogenesis and DNA damage repair are underway [24]. Furthermore, studies have identified a positive correlation between single gene perturbations on autosomal chromosomes and non-syndromic POI, implicating genes such as *growth differentiation factor 9 (GDF9)*, *folliculogenesis-specific basic helix–loop–helix (FIGLA)*, *NOBOX oogenesis homeobox (NOBOX)*, *estrogen receptor 1 (ESR1)*, *follicle-stimulating hormone receptor (FSHR)*, and *nanos homolog 3 (NANOS3)*.

Also, autosomal genes, including *B-cell lymphoma 2 (BCL2)*, *BCL2-associated X apoptosis regulator (BAX)*, and *cyclin-dependent kinase inhibitor 1B (DKN1B)*, have been examined as potential candidates [22]. It is noteworthy that the interplay among these genes might also play a role in the development of POI. However, it is important to mention that routine autosomal genetic testing is not currently recommended for patients with POI [2]. 

### 2.2. Autoimmune Ovarian Damage

Autoimmune diseases contribute to approximately 5–17% of POI cases [4]. Among these autoimmune conditions, autoimmune polyendocrine syndrome is particularly associated with POI, with a prevalence of 0.1%. This syndrome includes autoimmune thyroid disease, Addison’s disease, and type 1 diabetes mellitus, apart from POI. To screen for autoimmune disorder-induced POI, tests for thyroid peroxidase antibodies (TPO-Abs), adrenal 21-hydroxylase antibodies (21OH-Abs), and fasting glucose levels are recommended in cases of unknown origin, with adrenocortical antibodies (ACAs) being an alternative to 21OH-Abs. Positive 21OH-Ab/ACA test results warrant a referral to an endocrinologist for further investigation of adrenal disorders and Addison’s disease. Patients positive for TPO-Ab should undergo annual thyroid-stimulating hormone tests. Conversely, as the connection between POI and diabetes has not been thoroughly elucidated, routine screening for diabetes is not recommended [4]. If the 21OH-Ab/ACA and TPO-Ab test results are negative, no additional tests are deemed necessary [2].

### 2.3. Infectious Causes

There have been several previous reports of POI following infectious conditions, such as mumps, human immunodeficiency virus, herpes zoster, cytomegalovirus, tuberculosis, malaria, varicella, and shigella infections [2]. Only mumps oophoritis is considered a cause of POI [25]. Studies on other infections were limited to case reports with no statistical significance. POI caused by infection accounts for less than 1% of all cases. Therefore, screening for infections in patients with POI is unnecessary. 

### 2.4. Iatrogenic Causes

Iatrogenic POI has a prevalence ranging from 6% to 47% [4,26]. Notably, treatments such as chemotherapy and radiotherapy for malignancies or benign conditions contribute markedly to this disease. Ovarian surgery is another significant risk factor for iatrogenic POI, and a study by Ishizuka et al. indicated that POI resulting from ovarian surgery constitutes 64% of all iatrogenic cases [7]. Therefore, it is crucial to discuss the potential for iatrogenic POI with patients before they undergo gonadotoxic medical or surgical treatments. Moreover, when such interventions are deemed necessary, healthcare providers should actively present and consider fertility-sparing methods to mitigate their impact on patients’ reproductive capabilities. This proactive approach ensures that patients are well-informed and provided with options that align with their reproductive goals. 

### 2.5. Lifestyle Factors

Although body weight has been linked to various health issues, clinical evidence that supports the association between body weight and ovarian failure is limited. Several studies have identified a modest correlation between being underweight and experiencing an earlier onset of menopause, while being overweight is associated with a later age at menopause. The mechanism of decreased estrogen and leptin secretion in adipose tissue is proposed as a contributing factor [27]. A distinct mechanistic connection between excess weight and diminished reproductive function seems to exist. The correlation stems from the acknowledged impact of being overweight on elevated oxidative stress within the body, facilitated by various potential mechanisms, including heightened reactive oxygen species (ROS) production in adipose tissue [28,29]. Moreover, obesity is linked to persistent low-grade inflammation throughout the body [29]. Adipose tissue, functioning as a crucial endocrine organ, releases adipokines, which are believed to contribute to an inflammatory state. Implementing lifestyle changes, such as adopting a nutritious diet, engaging in regular physical activity, and achieving weight loss, can play a pivotal role in diminishing inflammation and alleviating the detrimental effects of health conditions caused by this specific etiology.

Although exercise is widely acknowledged for its positive impact on overall health and well-being, direct evidence linking it to the development of POI is lacking. However, there is limited evidence indicating a potential connection between exercise and the age at which women reach menopause. This relationship, though, appears to be intricate, with numerous other factors likely playing a role [30]. Some studies suggest that women who regularly engage in physical activity may undergo menopause at a later age compared to their sedentary counterparts [30]. Conversely, other studies have failed to establish a significant relationship [31]. The complex nature of this association underscores the need for further research to elucidate the nuanced interplay between exercise, menopausal age, and factors influencing reproductive health. 

Alcohol consumption, smoking, and chemicals (phthalates, bisphenols, and dioxins) have been suggested as causative factors [4]. Although cigarette smoking is associated with advancing the age of natural menopause, its relationship with POI has not been identified. Therefore, patients at risk of POI should be advised to quit smoking [2].

### 2.6. Environmental Causes

Exposure to toxic substances is a rare contributor to POI. These substances, whether natural or artificial, which interfere with the endocrine system, are known as endocrine-disrupting chemicals (EDCs). Numerous studies have examined the adverse effects of EDCs on *human* and animal reproductive functions, often resulting in infertility. Examples of EDCs include bisphenol A, phthalates, endosulfan and various organophosphates [32,33,34]. It is suggested that EDCs disrupt hormone signaling pathways, accelerate ovarian aging, and induce epigenetic changes [34]. However, the challenges in investigating the effects of environmental factors on POI lie in the limited detection methods available and the diverse sources of EDCs.

### 2.7. Human Papilloma Virus (HPV) Vaccination

Although POI following HPV vaccination has been reported in a few cases [35], a meta-analysis revealed that there was no significant risk of the disease in quadrivalent, bivalent, or 9-valent vaccinated patients compared to that in unvaccinated controls [36]. 

### 2.8. Coronavirus Disease 2019 (COVID-19) Infection and Vaccination

*Severe acute respiratory syndrome coronavirus 2 (SARS-CoV-2)* induces an immune reaction by binding to host cells, with angiotensin-converting enzyme (ACE2) serving as the primary receptor for the virus to enter cells. Upon invasion, *SARS-CoV-2* downregulates ACE2 expression, disrupts the renin–angiotensin system, and exacerbates the proinflammatory response through angiotensin-II [37]. ACE2 is highly expressed in female reproductive organs, including the ovaries, endometrium, and fallopian tubes, and plays a crucial role in the physiology of the ovaries and uterus [37]. Recent reports suggested that *COVID-19* can negatively impact female fertility potential, affecting granulosa cells and the ovarian tissue, potentially leading to poor oocyte quality and decreased ovarian function [38]. Additionally, the ACE2 distribution in the placenta and endometrium raises concerns about the potential hindrance of implantation and the increased risk of early abortion due to *COVID-19*. Conversely, Orvieto et al. [39] argued that *COVID-19* vaccination did not affect implantation rates or embryo quality in in vitro fertilization–embryo transfer patients. However, compared to healthy individuals, patients with *COVID-19* show a significant decrease in the implantation rate and embryo quality, potentially linked to a disrupted follicular microenvironment. A study by Bentov et al. [40] suggested that the quality of oocytes is similar between *COVID-19* vaccinated and unvaccinated groups of people. Consequently, while several studies have suggested that *COVID-19* infection may adversely affect oocyte or embryo quality, implantation, and pregnancy maintenance, reports on *COVID-19* vaccination have indicated no adverse impact on female fertility and pregnancy. 

## 3. Clinical Manifestations of POI

Patients with POI may present symptoms similar to those observed during natural menopause. Menstrual irregularity and infertility often occur several years before a diagnosis of POI [7]. In addition, POI can cause long-term sequelae, warranting the need for physicians to adequately inquire about the symptoms to avoid misdiagnosis and ensure optimal management. 

### 3.1. Menstrual Irregularity 

Menstrual cycle disturbance is the most common initial symptom of POI, with some patients consulting a physician for primary amenorrhea. However, no characteristic menstrual disturbance pattern could predict POI. Some patients suddenly experience sudden menstrual irregularity, while others experience prolonged or slow changes in oligomenorrhea [41,42]. Meanwhile, the use of oral contraceptives (OCs) can disguise the apparent features of POI. Therefore, if a woman fails to recover a regular menstrual cycle or conceives after ceasing OCs, POI should be suspected. 

### 3.2. Vasomotor Symptoms and Other Estrogen-Deficient Symptoms

Hypoestrogenism, a major characteristic of POI, can cause vasomotor symptoms, with the common ones being hot flashes and night sweats. Other estrogen-deficient symptoms include vaginal dryness, dyspareunia, mood changes, and poor concentration. The severity and duration of these symptoms vary depending on the fluctuations in ovarian endocrine function. Occasionally, symptoms can worsen or be alleviated [43,44]. Additionally, some patients with POI may not experience these symptoms. Patients with POI and primary amenorrhea are less likely to experience hypoestrogenic symptoms compared to those with surgically induced POI, who tend to have more severe symptoms [3].

### 3.3. Neurologic Symptoms

Several studies have explored the cognitive function in patients with POI, particularly in iatrogenic cases, where a sudden decline in verbal memory during the acute phase has been reported. Moreover, findings from various prospective and retrospective observational studies indicate that patients with surgical POI who do not undergo hormone replacement therapy exhibit impaired cognitive function and are at a higher risk of developing dementia and Parkinson’s disease [45,46,47], suggesting that early estrogen deficiency may negatively affect neurological function. Hence, it is advisable to consider hormone replacement therapy in patients with POI as a potential measure to reduce the risk of cognitive impairment, at least until the average natural menopausal age is reached. However, it is crucial to note that these findings are insufficient to draw a definitive conclusion and are primarily based on studies involving patients who underwent surgical oophorectomy [2,7].

### 3.4. Cardiovascular Diseases

Women with POI have an elevated risk of CVD and earlier onset of coronary heart disease, leading to increased mortality [48,49,50]. A cohort study reported an approximately 2-year shorter life expectancy in women with POI than in those experiencing menopause after the age of 55 years [51]. Multiple studies have elucidated the mechanisms underlying the increased risk of CVD in patients with POI. Factors such as impaired vascular endothelial function, autonomic dysfunction, abnormal lipid profiles, and increased incidence of metabolic syndrome are believed to contribute to CVD in these patients [52]. Notably, lifestyle factors such as obesity, smoking, and a lack of physical activity act as additional risk factors for CVD. Consequently, clinicians must inform patients with POI about the heightened risk of CVD and discuss potential ameliorating factors [2].

### 3.5. Bone Mineral Density (BMD)

Estrogen deficiency induced by premature ovarian insufficiency has a detrimental effect on bone health, decreasing the BMD and increasing the risk of fractures later in life [53]. The disruption of skeletal homeostasis, which is maintained by osteoblast progenitors and osteoclasts, results from estrogen loss [54]. Consequently, the BMD assessment is crucial for all patients upon POI diagnosis. Dual-energy X-ray absorptiometry (DEXA) is the gold standard for evaluating the BMD of the lumbar spine, hip, and forearm. If the BMD falls within the normal range and the patient is on sufficient systemic estrogen therapy, additional DEXA testing is typically not recommended. However, in patients with reduced BMD, a comprehensive examination of the estrogen treatment efficacy and other risk factors is necessary. If osteoporosis is confirmed, treatment options such as estrogen replacement therapy or alternative approaches are considered, with repetitive BMD measurements recommended within a 5-year interval for patients with osteoporotic POI [2,54].

The key risk factors contributing to the reduced BMD in POI include the severity and duration of the estrogen deficiency as well as poor compliance with estrogen replacement therapy. Additional risk factors include early diagnosis of POI, delayed diagnosis (more than 1 year), serum vitamin D levels below 32 ng/mL, low calcium intake, and lack of physical exercise [54]. Estrogen replacement therapy, primarily using OCs, is a common approach. Alternatively, medical treatments include bisphosphonates, selective estrogen receptor modulators, parathyroid hormone derivatives, and combined calcium and vitamin D supplements [2,54].

### 3.6. Infertility and Pregnancy in Patients with POI 

Patients with POI generally show a 5–10% natural conception rate [6]. Several methods, including estrogen therapy and administration of gonadotropins, corticosteroids, and immunosuppressants, have been used to increase the ovulation and pregnancy rates in these patients. However, these attempts neither improve ovarian function nor increase pregnancy rates [2,10]. At present, oocyte donation or adoption is the gold standard for improving pregnancy rates.

It is concerning that pregnancy in patients with POI is related to adverse obstetric outcomes. However, according to various case reports, spontaneous pregnancy in idiopathic POI or chemotherapy-induced POI does not show significant differences in obstetric or neonatal complications [10,55]. In contrast, pelvic irradiation increases the risk of miscarriage, premature birth, low birth weight, stillbirth, postpartum bleeding, and neonatal hemorrhage in pregnant women with POI [56,57]. 

## 4. Conventional Treatment of POI

Treatment of POI involves symptom alleviation, hormone replacement therapy, psychological support, and counseling for contraception and fertility. The treatment plan is determined based on the cause of POI, severity of symptoms, family heredity, genetic abnormalities, and underlying conditions.

### 4.1. Hormone Replacement Therapy

The benefits of hormone replacement therapy for patients with POI are well established; it provides relief from vasomotor and genitourinary symptoms, supports bone health, and improves sexual function. Although evidence regarding the impact of hormone replacement therapy on cardiovascular health is inconclusive, its use until natural menopause is generally recommended to reduce the risk of CVD. Hormone replacement therapy may also positively affect life expectancy, quality of life, and cognitive function; however, further evidence is required to confirm these findings. Consequently, women with POI without contraindications for hormone replacement therapy are advised to promptly consider its initiation [2,4]. Regarding the type and dosage of estrogen, 17β-estradiol is preferred to ethinyl estradiol or conjugated equine estrogen. The recommended oral doses include 1–2 mg daily of 17β-estradiol, 0.625–1.25 mg daily of conjugated equine estrogen, or 10 μg daily of ethinyl estrogen. Transdermal estradiol, particularly at a dose of 75–100 μg, is recommended for patients with POI who have specific conditions such as migraine with aura, hypertension, obesity, overweight, and an increased risk of venous thromboembolism (VTE). Transdermal estrogen is favored because it avoids first-pass metabolism in the liver, especially considering the long duration of estrogen use in patients with POI [4,7]. Before initiating hormone replacement therapy, it is essential to inform women with POI that hormone replacement therapy does not increase the risk of breast cancer before natural menopause. Patients with a history of VTE or thrombophilic disease should be referred to a hematologist. Hormone replacement therapy is contraindicated for breast cancer survivors with POI. In patients with intact uteri, progesterone should be combined with estrogen to prevent endometrial hyperplasia or cancer. Commonly prescribed options include oral micronized progesterone (100–200 mg for 12–14 days per month) or oral medroxyprogesterone acetate (2.5–5 mg daily or 10 mg daily for 12 days per month). During pregnancy, hormone replacement therapy is usually discontinued. The use of concurrent progesterone for women with POI without an intact uterus has yet to be established. Prompt initiation of hormone replacement therapy is crucial for the diagnosis of POI unless contraindicated. Annual monitoring for compliance and symptoms is recommended, and the estrogen dosage can be adjusted based on symptoms. Moreover, hormone replacement therapy is continued until the age of natural menopause [2,58,59,60].

### 4.2. Contraception

It has been reported that sporadic ovulation occurs in 23%–75% of women with POI; recovery of the menstrual cycle is observed in 25–50% of such patients [4,7]. Therefore, proper contraception in addition to hormone replacement therapy should be provided to patients with POI if they do not desire pregnancy. Combined oral contraceptives (OCs), levonorgestrel intrauterine device with estrogen replacement, or the barrier method are recommended [58,59,60]. According to the North American Menopause Society guidelines published in 2022, combined OCs may be an effective alternative to hormone replacement therapy, with additional contraceptive gain in healthy younger women [60].

### 4.3. Infertility Treatment

For most patients with POI, the significant impact on fertility often takes precedence, as many of these individuals aspire to have genetically inherited children. However, only a small percentage (approximately 5–10%) of patients are known to conceive naturally [61]. Oocyte donation is the current gold standard for treatment [2,4]. It is noteworthy that oocyte donation from sisters exhibits a higher cancellation rate and lower ovarian response than that from altruistic donors, likely due to the close genetic relevance among the siblings of patients with POI [62]. Hence, clinicians must communicate this information to patients when considering oocyte donation. Apart from adoption or oocyte donation, several less effective infertility treatments exist, including ovulation induction, use of gonadotropin or gonadotropin-releasing hormone agonists, estrogen priming, danazol, and corticosteroids [63]. However, none of these methods has demonstrated a clear advantage in increasing pregnancy rates or improving fertility compared to other methods [64]. A recent retrospective study reported promising outcomes of in vitro fertilization (IVF) and frozen embryo transfer in patients with POI. Patients who underwent repetitive oocyte retrieval to obtain autologous viable embryos showed a cumulative clinical pregnancy rate and cumulative live birth rate comparable to those in an age-matched normal control group [65]. It is noteworthy that this positive outcome is limited to women with POI with a substantial number of follicles, typically seen in cases caused by autoimmunity or in those diagnosed at an early age [4]. Therefore, there is an urgent need to develop innovative infertility treatment methods tailored to the unique challenges faced by patients with POI.

## 5. Current Stem Cell Therapy in POI

Stem cells (SCs) are undifferentiated or partially differentiated cells derived from embryos or adult tissues; they possess the unique ability to self-renew and differentiate into various cell types. They are broadly categorized as embryonic SCs (ESCs), adult SCs (ASCs), and induced pluripotent SCs (iPSCs) [66]. Mesenchymal SCs (MSCs) constitute a specific subset of ASCs that originate from various tissues, such as the bone marrow (BM), adipose tissue (AT), menstrual blood, umbilical cord (UC), amniotic fluid, amniotic membrane, placenta, and endometrium [55,67]. The medical application of SCs, known as stem cell therapy (SCT), leverages their inherent properties to restore function in damaged organs [67,68]. Specifically, SCT holds promise for the regeneration of ovarian function and oocytes in patients with POI. 

### 5.1. Mechanism of Stem Cell Therapy (SCT) in POI

SCs play a crucial therapeutic role in POI by engaging in various mechanisms (Figure 1). One important mechanism is homing, a physiological process by which circulating or extrinsic SCs navigate to their intended destinations. For instance, intravenously injected hematopoietic SCs and progenitor cells have been detected in the bone marrow. Although the homing process of MSCs is not fully understood, it is expected to reduce the number of cells required for successful transplantation [69].

In regenerative medicine, the paracrine action is considered the most critical mechanism of SCs. Paracrine effects involve the release of various cytokines that facilitate neovascularization, anti-inflammation, anti-apoptosis, anti-fibrosis, and immune regulation [70]. In the context of POI, it has been observed that the conditioned medium derived from SCs protects the ovary from age-related damage, emphasizing the pivotal role of the paracrine mechanism in improving ovarian function [71]. Specifically, studies have indicated that MSCs upregulate the release of hepatocyte growth factor (HGF), vascular endothelial cell growth factor (VEGF), insulin-like growth factor-1 (IGF-1), epidermal growth factor (EGF), fibroblast growth factor 2 (FGF2), granulocyte-colony stimulating factor (G-CSF), interleukin (IL)-8, IL-10, IL-11, and IL-15, while decreasing the secretion of tumor necrosis factor-alpha (TNFα) and IL-6 [72].

Furthermore, the therapeutic function of SCT is partially derived from exosomes, which are 30–160 nm sized vesicles attached to the cell membrane [67,73]. The therapeutic potential of exosomes, especially those derived from MSCs, for wound healing and repair is currently under investigation. These vesicles regulate intercellular communication through cell internalization, ligand–receptor interactions, and lipid membrane fusion, activating the signaling cascades involved in wound healing. Studies have revealed the activation of *protein kinase B (AKT)*, *signal transducer and activator of transcription 3 (STAT3)*, and extracellular signal-regulated kinase (ERK) by MSC-derived exosomes, along with an increase in the levels of growth factors such as HGF, IGF1, nerve growth factor (NGF), and stromal-derived growth factor-1 (SDF1) [74]. Given the potential of cell-free exosome therapy to mitigate immune rejection, vascular obstruction risk, and tumor mutation risk, it has better clinical value than conventional stem cell regimens [67]. Using SCs as a source of exosomes is another promising avenue in regenerative medicine. 

However, the effect of SC differentiation in patients with POI is yet to be established. It remains unknown whether MSCs or ovarian SCs (OSCs) can differentiate into oocytes [70,75]. However, animal studies have demonstrated the differentiation of MSCs into granulosa cells, indicating potential for restoring ovarian endocrine function and folliculogenesis in rodents [72,76]. The well-noted association between ovarian aging and mitochondrial dysfunction involves a significant reduction in the mitochondrial DNA (mtDNA) content in oocytes of women with POI compared to that in healthy women [77]. Conventional approaches using antioxidants to address mitochondrial damage in patients with POI have shown limited efficacy. SCT is an innovative approach to enhance oocyte quality by transporting mitochondria to adjacent cells through tunnel nanotubes. This transfer is facilitated by inflammatory cytokines secreted by SCs that promote tunnel nanotube formation. Wang et al. suggested that adipose tissue-derived SCs (ADSCs) can improve oocyte quality, embryo development, and fertility in elderly mice, indicating that mitochondrial transfer could be a promising option for enhancing oocyte quality in patients with POI [78]. 

### 5.2. Embryonic Stem Cells (ESCs)

ESCs are derived from the inner cell mass of the blastocyst, a developmental stage that occurs 3–6 days after the fertilization of *human* embryos and precedes uterine implantation. *Human* embryonic stem cells (*h*ESCs) can be obtained from supernumerary *human* IVF embryos from infertile couples [79,80] (Figure 2). ESCs have several advantages over other SC types. First, *h*ESCs are normal cells derived from *human* embryos, whereas iPSCs are generated from adult cells. Consequently, *h*ESCs are considered genetically stable. Furthermore, ESCs exhibit nearly infinite pluripotency, allowing limitless cell division. They maintain a normal karyotype during differentiation and can differentiate into all three germ layers (ectoderm, endoderm, and mesoderm). Lastly, *h*ESCs are less likely to induce immunogenic reactions and malignant changes compared to iPSCs [80]. Owing to these qualities, ESCs are regarded as a promising source in regenerative medicine.

Hübner et al. demonstrated that cultured mouse ESCs develop into oogonia. These oogonia undergo meiosis, recruit neighboring cells to form follicle-like structures, and develop into blastocysts [81]. Kehler et al. [82] and Kerkis et al. [83] demonstrated the differentiation of ESCs into primordial germ cells in mice and *humans*. In mice, the generated germ cells were capable of undergoing meiosis and forming both male and female gametes. The proper expression of various germ cell-specific genes, such as *Oct-4*, *Mvh*, *Stella*, *Dazl*, *Piwil 2*, *Pdrd 1*, *Rex 14*, *Rnf 17*, *Bmp8b*, *Acrosin, Stra-8*, *Haprin*, *LH-R*, *Gdf9*, *Zp3*, *Zp2*, S*ycp1*, and *Sycp3*, is essential for their differentiation into gamete-like cells [83]. Bahrehbar et al. [84] reported that ESC-derived MSCs restored hormone secretion, the survival rate, and reproductive function in a chemotherapy-induced POI mouse model. Notably, these results were comparable to those observed in the group treated with bone marrow-derived mesenchymal stem cells. However, ongoing trials involving ESCs in POI are limited because of ethical and technical challenges. Obtaining *h*ESCs from the inner cell mass of blastocysts often involves the destruction of embryos. A more innovative technique that allows the acquisition of *h*ESCs from a single embryo biopsy has been developed, thus addressing ethical concerns [85]. Despite their infinite pluripotency, the prolonged in vitro culture of *h*ESCs can lead to genetic damage and potential tumor changes. Although whole-genome sequencing is an ideal method to exclude cancerous cells, it is time-consuming and expensive for routine use. Alternatively, single-nucleotide polymorphism arrays and karyotyping can be employed to assess the genetic stability of *h*ESCs [80,86,87]. Furthermore, *h*ESCs have the potential risk of immune rejection because they originate from allogeneic sources. 

### 5.3. Induced Pluripotent Stem Cells (iPSCs) 

Induced pluripotent stem cells (iPSCs) were first introduced in 2006 by Yamanaka and Takahashi [88]. They successfully induced PSCs from mouse fibroblasts using four transcription factors, *Oct4*, *klf4*, *sox2*, and *c-myc*, collectively known as *OSKM* (Figure 2). These genes can be used to reprogram somatic cells into iPSCs, making them a prospective tool for SCT. iPSCs address ethical concerns associated with ESCs as they are derived from somatic cells. Furthermore, iPSCs share similarities with ESCs in terms of the morphology, expression of cell surface markers, telomerase activity, capacity to differentiate into all three lineages, and maintenance of a normal karyotype [89]. Given that iPSCs originate from adult somatic tissue, they are readily available and evoke fewer immune reactions, providing a solution to the challenge of finding immunocompetent SCs due to the complexity of the *human* immune system. 

Autologous iPSCs hold promise in overcoming immune compatibility issues [90,91]. In 2013, Liu et al. [92] reported the in vitro differentiation of *human* induced pluripotent stem cells (*h*iPSCs) into hormone-sensitive ovarian epithelial (OSE)-like cells under the influence of microRNA-17-3p (miR-17-3p). miR-17-3p suppresses vimentin expression, leading to the morphological transition of iPSCs into fibroblast-like cells. Injecting these OSE-like cells into the ovaries of a POI-induced mouse revealed increased expressions of the cytokeratin 7 and ERβ proteins, decreased fibronectin and vimentin, and increased ovarian weight and plasma estrogen (E2) levels. Yamashiro et al. succeeded in generating *human* oogonia from iPSCs in vitro [93,94]. After inducing hiPSCs into *human* primordial germ cell-like cells, they differentiated into oogonia-like cells during a 4-month culture. This in vitro culture was conducted in a xenogeneic reconstituted ovary with mouse embryonic ovarian somatic cells.

Despite these promising results, it is crucial to note that iPSCs carry neoplastic potential, as oncogenes, such as c-myc, are utilized in the reprogramming process. In addition, residual epigenetic imprints, gene silencing, and genomic instability may persist. The development of efficient and safe protocols is paramount because current hiPSC protocols are time-consuming and expensive. Consequently, iPSC transplantation in *humans* has not yet been widely adopted.

### 5.4. MSCs

MSCs, which are characterized by adherence to plastic surfaces and a fibroblast-like morphology [95], are found in various sources, such as the bone marrow, AT, peripheral blood, menstrual blood, endometrium, heart, and muscles. Perinatal sources of MSCs include the placenta, UC, cord blood, Wharton’s jelly, and placental or amnion membrane [96]. Adult tissue-derived MSCs often require invasive cell collection. However, these cells offer advantages in terms of autologous immunity. Meanwhile, MSCs of perinatal origin hardly pose ethical issues, as their sources are generally considered waste and destined to be discarded.

**Figure 2 biomolecules-14-00242-f002:**
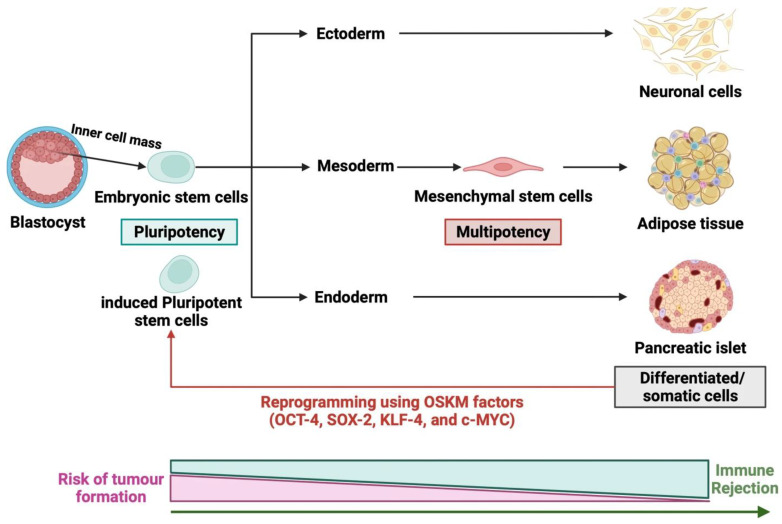
Various sources of stem cells. (1) *Human* pluripotent stem cells, including embryonic stem cells and induced pluripotent stem cells, exhibit pluripotency and can differentiate into other cell types, including ectoderm, mesoderm, and endoderm. For the reprogramming of induced pluripotent stem cells from somatic cells, *OSKM* factors are utilized. (2) Mesenchymal stem cells are derived from mesoderm. They exhibit multipotency, self-renewal ability in vitro and differentiation potential into mesenchymal lineages. Reprinted with permission from Ref. [96]. Copyright 2022 Springer Nature.

#### 5.4.1. Bone Marrow-Derived Mesenchymal Stem Cells (BMSCs)

Bone marrow-derived mesenchymal stem cells are mononuclear cells comprising mesenchymal stem cells, hematopoietic cells, and endothelial progenitor cells. Conventionally, specimens are collected from the pelvis, femur, and sternum using aspiration needles. However, this invasive method poses an obstacle to the widespread clinical application of bone marrow-derived mesenchymal stem cells. Recently, bone marrow-derived mesenchymal stem cells were easily obtained from peripheral blood after stimulation with G-CSF.

Since Lee et al. [97] reported that bone marrow transplantation restored ovarian function in a chemotherapy-induced POI *murine* model, bone marrow-derived mesenchymal stem cells have been regarded as a promising treatment option for infertility. In 2008, Fu et al. [98] first applied BMSCs in a *murine* model of chemotherapy-induced POI to investigate their regenerative effects. To date, several studies on BMSCs have reported promising results, including an increased ovarian volume [99,100], elevated E2 levels [99,101,102,103], restoration of the menstrual cycle [99], improved menopausal symptoms [99], increased anti-Mullerian hormone (AMH) levels [100,104], an increased number of antral follicles [101,102,103,105,106], higher pregnancy rates [97,100,101,105], and decreased apoptosis [98,101,107]. Chemotherapy-induced POI animal models, such as mice [97,100,101], rats [98,102,106,107], and rabbits [103], have been used to evaluate the effects of BMSCs. Three studies were conducted in *humans* [99,104,105]. One study reported a live birth after SCT [104], and another study revealed five pregnancies [105]. None of these *human* studies reported any adverse events.

However, the mechanism of action of bone marrow-derived mesenchymal stem cells in POI remains under investigation. Current hypotheses include paracrine action, reduced apoptosis, increased angiogenesis, and direct differentiation. A growing body of evidence suggests that *human* bone marrow-derived mesenchymal stem cells regulate cytokine secretion from Th1 and Th2 cells [108]. Additionally, increased secretion of secretomes [99], VEGF [103], *Bcl-2* [98], and miR-21 [107], as well as downregulation of *PTEN* and *PDCD4* [107], have been observed. These studies suggested that BMSCs regulate autocrine or paracrine environment, thereby reducing apoptosis and facilitating angiogenesis. In contrast, a study using a rabbit model revealed the direct differentiation of BMSCs into ovarian tissue cells [103]. However, differentiation into oocytes or primordial follicles was not observed.

#### 5.4.2. Menstrual Blood Mesenchymal Stem Cells (MenSCs)

MenSCs, also known as endometrial stem cells, were identified in 2007. On day 2 of the menstrual cycle, menstrual blood is collected from healthy women using menstrual or urine cups. Once a sufficient volume of blood is collected, it is treated with antibiotics such as amphotericin B, streptomycin, and penicillin. Subsequently, disodium ethylenediaminetetraacetic acid (EDTA-Na2), phosphate-buffered saline (PBS), and Ficoll-Paque are added to the samples. The sample is centrifuged, and the middle layer is cultured in a medium [109]. MenSCs exhibit high proliferation rates, pluripotency, and low immunogenicity. They are easily obtained and serve as allogeneic or autologous SC sources for women with POI without causing pain or ethical concerns [110].

The postulated therapeutic mechanisms of MenSCs include differentiation into the target tissue [111], immunoregulation [112], paracrine effects [112,113,114], and homing and engraftment [111,112,113,114]. In a study by Wang et al. [112], MenSCs or MenSC-derived media were intravenously injected into a POI mouse model, resulting in reduced apoptosis in granulosa cells, decreased fibrosis of the ovarian interstitium, an increased number of follicles, and higher serum E2 levels. The authors suggested that secretion of FGF2 partially restores damaged ovaries. In another study, *human* MenSCs (*h*MenSCs) were injected into the ovaries of a POI mouse model, revealing an increased number of follicles, elevated plasma E2 levels, and heightened expression of AMH, inhibin α/β, FSH receptor, and antigen Kiel 67 (Ki 67) in ovarian tissue. Lai et al. [111] reported the differentiation of transplanted *h*MenSCs by observing the migration and localization of green fluorescent protein-labeled *h*MenSCs in the ovarian stroma 48 h after transplantation. Moreover, a *human* clinical trial has demonstrated promising results [115]. Autologous *h*MenSCs were injected into the left ovary of women with POI, resulting in natural pregnancies in 4 of 15 women 3 months after transplantation. The control group that underwent routine intracytoplasmic sperm injection did not achieve pregnancy during the same period. Clinical pregnancies were observed in 7 of 15 women in the experimental group, with 5 of 7 successfully giving birth. No significant differences in the AMH levels, mean antral follicle count (AFC), or oocyte number were observed between the two groups. However, the oocyte fertilization rate and embryo number improved in the *h*MenSC group, suggesting paracrine effects rather than oocyte differentiation. 

The challenges in *h*MenSCs therapy include determining the duration of MenSC survival in foreign bodies and investigating the long-term safety of MenSCs in *humans*. These aspects and considerations require further research.

#### 5.4.3. Adipose Tissue-Derived Stem Cells (ADSCs) 

ADSCs can be extracted from the fat tissue during arm, thigh, or abdominal liposuction, as well as during abdominal plastic surgery. They are not only easily obtainable but also abundant. Additionally, the high proportion of *human* ADSCs (*h*ADSCs) in the AT eliminates the need for a prolonged in vitro culture [116,117]. Neri et al. [118] demonstrated the genetic stability and replicative senescence of ADSCs in culture.

Various studies have explored the mechanisms of action of *h*ADSCs in POI, with paracrine regulation being frequently postulated [72,119,120]. These investigations suggest that downregulation of the *phosphoinositide 3-kinases (PI3K)*/*protein kinase B (Akt)*/mammalian target of rapamycin (mTOR) axis and SMAD pathways contributes to the repair of chemotherapy-induced POI in ovaries. Takehara et al. [72] reported increased angiogenesis in rats transplanted with ADSCs. Specifically, VEGF, IGF-1, and HGF induced neovascularization. Sun et al. [121] have refuted the differentiation of ADSCs into ovarian tissue cells. The authors demonstrated altered gene expression related to follicle formation and ovulation, supporting the paracrine effects of ADSCs, as previously discussed in other studies.

Despite their accessibility and safety, there are currently only a limited number of studies on ADSCs in POI, most of which have been conducted using *murine* models. Further basic human studies are required to broaden their application.

#### 5.4.4. Umbilical Cord-Derived Mesenchymal Stem Cells (UC-MSCs)

UC-MSCs can be isolated from the entire UC or specific segments, such as Wharton’s jelly, a gelatinous connective tissue within the UC. Once the vessels are removed, Wharton’s jelly is cut into small pieces to facilitate SC culture, resulting in Wharton’s jelly-derived mesenchymal stem cells (WJ-MSCs). These UC-MSCs, similar to other MSCs, exhibit multipotency, self-renewal ability, and the capacity to differentiate into three lineages. UC-MSCs, particularly WJ-MSCs, demonstrate faster proliferation rates and greater expansion capabilities than those of other adult MSCs, such as ADSCs and bone marrow-derived stem cells (BMSCs) [96,122]. In addition, UC-MSCs, specifically WJ-MSCs, express the surface markers CD29, CD44, CD73, CD90, and CD105 but lack CD31, CD45, and *HLA-DR85* [89]. Ding et al. [123] reported various improvements in clinical indicators following the transplantation of UC-MSCs via a collagen scaffold in *humans*. These improvements included increased serum E2 levels, enhanced follicle development, increased follicle count, and successful pregnancies. Activation of primordial follicles has been suggested to occur through the phosphorylation of forkhead box O3 (FOXO3a) and forkhead box protein O1 (FOXO1). Studies by Li et al. [124] and Zhu et al. [125] using *murine* models of POI have indicated that UC-MSCs exert a paracrine effect, facilitating anti-apoptosis and cytokine secretion. Zhu et al. further compared the efficacy of intraovarian injection of UC-MSCs with intravenous injection, noting faster restoration of ovarian function in the intraovarian injection group, although the long-term restoration was similar in both groups [125]. In contrast, Wang et al. [126] investigated the differentiation of UC-MSCs and argued that transcriptional regulation, the G-protein-coupled receptor protein signaling pathway, mitogen-activated protein kinase (MAPK) pathway, and the insulin pathway are involved in paracrine action.

#### 5.4.5. Amniotic Mesenchymal Stem Cells (AMSCs)

AMSCs include amniotic epithelial cells, amniotic fluid stem cells, and amnion-derived stem cells. During embryogenesis, the amnion originates from the embryonic mesoderm and expresses the characteristic embryonic SC markers *Oct-4* and *SSEA-4* [127]. The specific cell surface markers of AMSCs include CD105, CD29, CD44, CD73, CD49d, and CD90 [128]. AMSCs exhibit high proliferative and self-renewal abilities. These characteristics suggest that in patients with POI, AMSCs exert paracrine activity, secrete exosomes, undergo differentiation, demonstrate homing, and exhibit anti-apoptotic and anti-fibrotic effects [76,127,129,130,131]. Zhu et al. [76] and Lai et al. [130] reported the differentiation of *human* amniotic epithelial cells and *human* amniotic fluid SCs into granulosa cells. This differentiation is more frequently observed in AMSCs than in other types of SCs. The primary functions of AMSCs are attributed to their paracrine activity, in which mediators such as EGF, HGF, VEGF, IGF-1, and exosomes are upregulated. These molecules play regulatory roles in angiogenesis, apoptosis, the cell cycle, and immune responses. 

#### 5.4.6. Placenta-Derived Mesenchymal Stem Cells (PMSCs)

The placenta functions as an organ that produces hormones and facilitates the exchange of nutrients, gases, and waste between the mother and the fetus. Placenta formation occurs shortly after embryo implantation. Given that the placenta is typically discarded after childbirth, obtaining PMSCs is relatively straightforward. Similar to other MSCs from perinatal sources, PMSCs are not associated with pain, invasive procedures, or ethical concerns. PMSCs exhibit high proliferative activity and can differentiate into various cell types, including osteoblasts, smooth muscle cells, adipocytes, endodermic pancreatic islet cells, liver cells, astrocytes, and ectodermic neurons [132]. Yin et al. [133] demonstrated that ovarian function was restored in a *murine* model of POI. This study was unique because POI was induced by the zona pellucida glycoprotein 3 (pAP3), creating an autoimmune POI model instead of using chemotherapy. The restoration of ovarian function and reduction in granulosa cell apoptosis were attributed to the involvement of regulatory T cells. Additionally, the paracrine action of transforming growth factor beta (TGF-β) and the increase in interferon-gamma (IFN-γ) levels were demonstrated. Despite these promising results, further investigations are required to assess the efficacy of PMSCs in *humans*. Table 1 summarizes the current and ongoing MSC therapies for treating patients with POI.

### 5.5. OSCs

Conventionally, a fixed primordial follicle pool and oocytes are thought to exist. According to this theory, oocytes are not generated after birth, and once menarche begins, primordial follicles and oocytes are depleted until menopause. Johnson et al. [136] challenged this hypothesis by reporting the presence of proliferative germ cells. These cells produce oocytes and follicles in the postnatal mammalian ovary and are termed OSCs, which have the potential to revolutionize infertility treatment.

In 2017, Liu et al. [137] isolated OSCs from neonatal mice and presented them as a string-like formation comprising an average of 15 cells. The authors observed that OSCs underwent vigorous mitosis. In addition, evidence supports the existence of OSCs in adults [138]. Putative ovarian MSCs were obtained from the ovarian cortex of adult *humans*. These cells express characteristic genes related to MSCs, such as CD105, CD44, CD90, M-CAM, CD73, VCAM1, and STRO-1. A recent study on OSCs reported intraovarian injection of female germline SCs into *murine* models, resulting in the differentiation of SCs into early stage oocytes, with a remarkable outcome in terms of offspring generation [135]. However, the existence of OSCs in *humans* and their characteristics and methods of isolation and culture remain challenging.

### 5.6. Limitations of Current SCT in the Clinical Setting of POI

Recently, significant progress has been made in understanding SCs’ mechanisms and their clinical applications. SCs from diverse sources are currently being investigated in animal and *human* studies. However, it is crucial to recognize that SCs can potentially become malignant. This is particularly pertinent in iPSCs, where several oncogenes for induction have raised concerns. The accumulation of genomic and epigenetic alterations poses a risk for tumor changes. Moreover, ethical considerations arise with using *h*ESCs, allogeneic SCs, and xenogeneic SCs. Another challenge for widespread implementation is the low productivity and high variability in cell quality. Addressing these obstacles may involve developing culture protocols and advancing cell-free SCT. 

Currently, studies on SCT in animal models are limited. In all the animal studies, POI was induced through chemotherapy, albeit with variations in the dosage, route, period, and type. However, these models may not fully reflect idiopathic or autoimmune POI in the ovaries, imposing limitations on extrapolating the findings to actual POI in women. Therefore, the mechanisms of SCT identified in the POI models may differ from those applicable to women with actual POI. Further studies exploring the various causes of POI in women could lead to breakthroughs in understanding and treating this condition.

## 6. Clinical Trials Involving SCTs for POI 

Numerous clinical trials investigating SCTs for POI have been documented at ClinicalTrials.gov, totaling 28 registered studies as of 15 December 2023. Within this cohort, two studies are suspended, one withdrawn, and another terminated. Additionally, the status of ten trials remains unknown, and two are deemed irrelevant. Among the twelve trials dedicated explicitly to SCT for POI, seven have concluded, two are actively enrolling participants, and two are in the pre-recruitment phase. Furthermore, one trial is active but not currently recruiting. Table 2 summarizes the current and ongoing clinical trials focusing on the use of SCs in the treatment of POI. 

To specify, two studies employ BMSCs as therapeutic tools for treating POI [139,140]. In both cases, autologous bone marrow was aspirated, and the obtained BMSCs were surgically transplanted into the ovaries. Particularly noteworthy is the study by Edessy et al. [139], who reported a successful live birth after treatment. Another study (NCT06132542) focused on ADSCs, although it is not currently recruiting participants. This phase 1 study is scheduled to commence recruitment on 15 January 2024 [141].

Several research groups are actively exploring the use of UC-MSCs. An ongoing trial (NCT05308342) involves the transplantation of *human* UC-MSCs into the ovaries of women with POI, followed by hormone replacement therapy [142]. In this study, the control group receives hormone replacement therapy only, with the primary outcome being the follicular development rate in the 9–12-month post-transplantation period. Another trial (NCT05138367) includes two intervention arms, one utilizing UC artery-derived perivascular SCs and the other using WJ-MSCs [143]. Both groups have undergone hormone replacement therapy, with the investigators focusing on changes in the blood flow as an outcome. While the phase 1 trials of the study have been completed, the results are pending.

Of interest is a trial (NCT02644447) investigating the efficacy of administration routes, comparing a group receiving *human* UC-MSC injections into the ovaries with a group receiving an injectable scaffold [144]. Another trial (NCT05494723) conducted by Bright Cell, Inc. is awaiting recruitment [145]. In this study, the investigators have extracted a drug named YB1113 from *h*UC-MSCs. This phase I trial aims to assess serum AMH, FSH, and E2 levels and AFC numbers in both the low-dose YB-1113 and high-dose YB-1113 groups while evaluating the treatment-related adverse events.

Cell-free therapy using *h*PMSCs is under development in a trial (NCT06072794) funded by the industry [146]. In this study, patients with POI receive exosome (EV-Pure™) injections, and the evaluation includes assessing the occurrence of adverse events and monitoring the serum levels of AMH, FSH, E2, and AFC. Another study (NCT01702935) focusing on OSCs is a completed prospective, observational study designed to uncover the existence of OSCs, with the results not yet disclosed [147].

In addition to conventional SCT, other novel approaches are currently under development. For instance, the four-step Autologous Ovarian Stem Cell Transplantation (ASCOT) trial (NCT04475744) involved the injection of autologous BMSCs along with G-CSF and activated platelet-rich plasma (PRP) into the ovaries [148]. The control group did not receive any intervention. This ongoing phase III trial is currently active but has not recruited participants. Similarly, another phase IV study (NCT02783937) involved the random allocation of a control group receiving subcutaneous saline injections and an experimental group treated with G-CSF twice daily for 5 days [149]. This study is unique in that it used a subcutaneous route rather than the more traditional intravenous or intraovarian route. Given the enhanced accessibility of G-CSF compared with that of SCT, these studies may provide valuable insights for the treatment of patients with POI in the future.

**Table 2 biomolecules-14-00242-t002:** Clinical trials of stem cell therapy for treating POI.

Identifier	Title	Site	Phase	Status	Source of Cells	Interventions	Outcome	Reference
NCT02151890	Pregnancy after stem cell transplantation in Premature Ovarian Failure	Al Azhar University, Egypt	Phase 1 Phase 2	Completed	SCs	Laparoscopic injection of SC sample in the ovaries of POI	No results posted	[150]
NCT02372474	“It is a Real” The First Baby Of Autologous Stem Cell Therapy in Premature Ovarian Failure	Al Azhar University, Egypt	Phase 1 Phase 2	Completed	BMSCs (Autologous)	BMSCs collected from iliac crest were injected laparoscopically	Successful live birth	[139]
NCT05308342	Clinical Study of *Human* Umbilical Cord Mesenchymal Stem Cells in the Treatment of Premature Ovarian Insufficiency	The affiliated Drum Towel Hospital of Nanjing University Medical School, China	Not Applicable	Recruiting	*h*UC-MSCs	Intraovarian injection of *h*UC-MSCs in POI women	No results posted	[142]
NCT06072794	A Proof of Concept Study to Evaluate Exosomes From *Human* Mesenchymal Stem Cells in Women With Premature Ovarian Insufficiency (POI)	Optimal Health Associates, United States	Phase 1	Recruiting	*h*PMSCs	Exosomes from *h*PD-MSCs are intravenously injected to POI women	No results posted	[146]
NCT02696889	Rejuvenation of Premature Ovarian Failure With Stem Cells	University of Illinois at Chicago, United States	Not Applicable	Completed	BMSCs (Autologous)	After bone marrow aspiration under anesthesia, obtained BMSCs are laparoscopically injected into the ovaries of POI women	No results posted	[140]
NCT06132542	Autologous ADMSC Transplantation in Patients With POI	Mongolian National University of Medical Science, Mongolia	Phase 1	Not yet recruiting	AD-MSCs (Autologous)	Ovarian injection of 5 million ADSCs	No results posted	[141]
NCT05138367	Effects of UCA-PSCs in Women With POF	Nanjing University, China	Phase 1	Completed	UC blood-MSCs Wharton’s jelly-MSCs	Group 1: Ovarian injection of UCA-PSC followed by HRT Group 2: Ovarian injection of WJ-MSC followed by HRT	No results posted	[143]
NCT02644447	Transplantation of HUC-MSCs With Injectable Collagen Scaffold for POF	The Affiliated Nanjing Drum Tower Hospital of Nanjing University Medical School, China	Phase 1 Phase 2	Completed	*h*UC-MSCs (Allogenic)	Group 1: Ovarian injection of 10 million allogeneic *h*UC-MSCs Group 2: Ovarian injection of 10 million allogeneic *h*UC-MSCs with injectable collagen scaffold	No results posted	[144]
NCT04475744	4-step ASCOT in POI Women to Promote Follicular Rescue	Instituto Valenciano de Infertilidad, IVI VALENCIA, Spain	Phase 3	Active, not recruiting	G-CSF PRP	Ovarian injection of G-CSF activated PRP	No results posted	[148]
NCT02783937	Filgrastim for Premature Ovarian Insufficiency	South Valley University, Egypt	Phase 4	Completed	G-CSF	Subcutaneous injection of G-CSF (30 million IU/ml) twice a day for five days	No results posted	[149]
NCT05494723	Safety and Efficacy of YB-1113 in Treatment of POI	Bright Cell, Inc., Irvine, CA, USA	Phase 1	Not yet recruiting	YB-1113 from *h*UC-MSCs	Intravenous infusion of low-dose or high-dose YB-1113	No results posted	[145]
NCT01702935	Ovarian Stem Cells From Women With Ovarian Insufficiency	National Institutes of Health Clinical Center, United States	Not applicable	Completed	OSCs	(1) Ovarian biopsy by laparoscopy or clinically indicated abdominal surgery in POI women (2) Isolation of oogonial stem cells in laboratory	No results posted	[147]

SCs Stem cells; BMSCs Bone marrow-derived mesenchymal stem cells; *h*UC-MSCs *Human* umbilical cord-derived mesenchymal stem cells; *h*PMSCs *Human* placenta-derived mesenchymal stem cells; *h*PD-MSCs *Human* placenta-derived mesenchymal stem cells; BM Bone marrow; POI Premature ovarian insufficiency; UCA-PSC Umbilical cord artery-perivascular stem cells; POF Premature ovarian failure; AD-MSCs Adipose tissue-derived mesenchymal stem cells; ADSCs Adipose tissue-derived mesenchymal stem cells; UC Umbilical cord; MSCs Mesenchymal stem cells; HRT Hormone replacement therapy; ASCOT Autologous stem cell ovarian transplantation; G-CSF Granulocyte colony stimulating factor; PRP Platelet-rich plasma; OSCs Ovarian stem cells.

## 7. Alternative Approaches for Infertile Patients with POI 

### 7.1. Ovarian Tissue Cryopreservation (OTC) and Autotransplantation

In 1999, Oktay et al. pioneered OTC and autotransplantation to restore ovarian endocrine function [151]. Donnez achieved the first successful live birth through OTC in 2004 [152]. OTC has transitioned from an experimental category to a clinically recognized method for fertility preservation endorsed by the American Society of Reproductive Medicine. Currently, OTC is used for patients with cancer seeking to safeguard their fertility before treatment or for those experiencing hormone replacement therapy-irresponsive menopausal symptoms. The global cumulative live birth rate per woman is 40%, prompting discussions on “elective” OTC for delaying natural menopause and extending childbearing age [153].

The OTC procedure commences with the surgical extraction of a portion, a unilateral ovary, or the entire ovary. Robotic surgery enhances the efficiency of OTC. After harvesting, the tissues undergo slow freezing and are stored in liquid nitrogen until transplantation. Thawed ovarian tissue is transplanted orthotopically (pelvic) or heterotopically (outside the pelvis). Studies on the xenografting of sheep and *human* ovarian tissues indicate that the loss of the primordial follicle reserve during slow freezing and rapid thawing is less than 10%. However, in most cases, follicular loss occurs after transplantation. During revascularization, a significant number of follicles suffer ischemic damage, leaving only 25% of the primordial follicles functional. This negative balance poses a major obstacle to elective OTC in healthy women with POF, necessitating advances in vascularization-enhancing approaches.

In conventional protocols, grafted ovarian cortex strips are attached to pelvic vascular structures for spontaneous vascularization, a process that lasts for 10 days. However, this duration exposes the primordial follicles to ischemic damage. Efforts to overcome this challenge involve trials with SCs, growth factors, basic FGF, VEGF, antioxidants, and androgens. Notably, one study revealed that ADSCs positively impacted the revascularization of xenografted ovarian tissue and follicle survival rates. The ADSC-treated group exhibited higher oxygenation, increased vascularization of the ovarian tissue, and enhanced follicle survival rates compared to those in the non-ADSC group [154].

Sphingosine-1-phosphate (S1P), an inhibitor of the ceramide-induced death pathway, exhibits the capacity to shield *human* ovarian primordial follicles from chemotherapy-induced apoptosis [89,155]. Notably, S1P stimulates endothelial cell migration. In a recent study, S1P was injected into autotransplanted ovaries, resulting in a remarkable reduction in the revascularization period from 10 days to 2–3 days. Moreover, patients who received S1P injections displayed a two-fold increase in microvascular density by day 10. However, the use of FTY-720, a clinically approved synthetic analog of S1P, has yielded contradictory results. The diminished revascularization observed with FTY-720 may be linked to receptor downregulation at high doses. Given that S1P has not yet been approved for clinical use and FTY-720 has demonstrated contradictory outcomes, further research is imperative [156].

Among patients undergoing OTC and autotransplantation for fertility preservation, at least two-thirds exhibited menopause reversal. In a preliminary study, 100% menopausal reversals were reported. A meta-analysis of 309 cases of OTC confirmed the restoration of ovarian endocrine function, as characterized by cyclic estradiol production and follicular development, in the majority of patients [157]. Over 40% of these patients successfully delivered at least one child after autotransplantation. The average duration of ovarian function post-transplantation was 26.9 months (range: 4–144 months), with an average age of 29.3 years (range: 9–44 years). Nevertheless, the elective use of OTC raises concerns about advancing the age of natural menopause, with the odds ratios ranging from 1.17 to 6.3. Currently, there is no evidence supporting the optimal amount of harvested ovarian tissue for elective storage. Another limitation in the context of POI is the average maintenance period of the grafted tissue, which stands at 26.9 months. Although sufficient for infertility treatment, it falls short of being an ideal alternative to hormone replacement therapy. In conclusion, OTC combined with SCT has the potential to enhance fertility in patients with POIs. Future studies focusing on OTC in patients with POI or in healthy women will be indispensable for establishing an optimal protocol.

### 7.2. Intraovarian Injection of PRP

PRP, which is derived from centrifuged autologous peripheral blood, is a novel approach for treating POI. Although various centrifugation protocols exist, with speeds ranging from 100 to 1350 Gay (G) or 1000–3200 rotations/min and durations spanning 4–20 min for the first centrifugation and 9–23 min for the second [158], the optimal preparation protocol remains undetermined. Notably, the platelet concentration in PRP is approximately 10 times higher than that in circulating blood [159].

PRP has several advantages. Given that it is of autologous origin, it is less likely to trigger immune reactions. Its high accessibility is attributed to its relatively simple and quick preparation using many commercially available kits. Despite widespread clinical use of PRP in dentistry, orthopedics, anesthesiology, and ophthalmology [160], the mechanism underlying the restoration of ovarian function remains elusive. Studies have suggested that PRP enhances the viability of primary and preantral follicles [161], as evidenced by the improved AMH levels after treatment, reflecting the pool of preantral follicles. Another proposed mechanism is restoration of the ovarian microenvironment, as marked by reduced oxidative stress, increased angiogenesis, and altered cytokine secretion [67,160]. These effects are attributed to the abundant platelets, growth factors, and alpha-granules present in PRP, releasing factors like TGF-β, VEGF, PDGF, IGF-1, and FGF-β [162,163].

Several studies have shown the positive effects of PRP in patients with infertility. Farimani et al. [164] reported improved oocyte quantity and quality in poor ovarian responders (PORs), with a pregnancy rate of 14.6%. A prospective study on POR demonstrated statistically significant improvements in the AMH, AFC, and FSH levels, with a pregnancy rate of 20.6% [165]. Interestingly, the age of 40 years was identified as the cut-off for PRP treatment. In addition to hormonal changes and pregnancy rates, one report showed that PRP increased the euploidy rate [166]. Despite these promising findings, challenges exist in the application of PRP in patients with POI, as studies have been predominantly conducted on PORs and discrepancies in the preparation protocols, timing, site, and amount of injection persist. Further studies exploring the various causes of POI are necessary to provide a more comprehensive understanding of the efficacy of PRP in this context. 

## 8. Conclusions

There is increasing optimism regarding SCT for infertility. Despite rapid advancements in assisted reproductive techniques over the past few decades, POI remains unresolved. SCs, with their pluripotency, self-renewal ability, and capacity to differentiate into various cell types, are being actively investigated for their clinical potential. Presently, the primary mechanisms of SCs in patients with POI involve paracrine effects, such as angiogenesis, anti-apoptosis, anti-fibrosis, and immunomodulation. In addition, several studies have reported differentiation into ovarian tissue cells. Progress has been made in understanding female germline cells. Although the fixed primordial follicle pool theory has been upheld for decades, recent studies have confirmed the presence of ovarian SCs. However, a comprehensive understanding of the mechanisms of SCs is crucial for their widespread clinical application. Unveiling homing mechanisms, developing less time-consuming culture protocols, reducing immune responses and the risk of tumor changes, and achieving uniform production of SCs are essential for the development of effective treatment approaches.

## Figures and Tables

**Figure 1 biomolecules-14-00242-f001:**
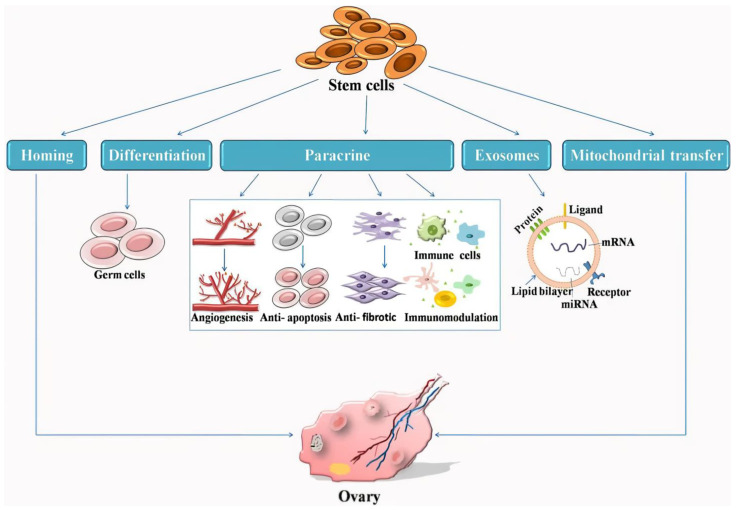
Schematic mechanisms of stem cell therapy in premature ovarian insufficiency. Homing is a vital process in which stem cells navigate to their intended destination, a critical aspect of the successful and efficient transplantation of stem cells. The differentiation of stem cells into various cell types, such as oocytes or female and male germline cells, can alter infertility treatment methods. In stem cell therapy, the paracrine effect is paramount in influencing angiogenesis, cellular apoptosis, scar fibrosis, and immune reactions. Exosomes, vesicles ranging from 30 to 160 nm in size and derived from stem cells, play a pivotal role in modulating intercellular communication. The replacement of damaged mitochondria in neighboring oocyte cells has the potential to enhance oocyte quality. Reprinted with permission from Ref. [67]. Copyright 2022 Springer Nature.

**Table 1 biomolecules-14-00242-t001:** Summary of current mesenchymal stem cell therapies for treating POI.

Type of Stem Cell	Species	Mechanisms	Route of Administration	Outcome	References
*h*ESCs	*Murine*	Paracrine effect Improved ovarian cell survival	Intraovarian	Similar result with BMSCs Restored hormone secretion, ovarian cell survival rate and reproductive function Generation of offspring	[84]
hiPSCs	*Murine*	miR-17-3p-induced differentiation of iPSCs to OSE-like cells	Intraovarian	Increased expressions of cytokeratin 7 and ERβ proteins Downregulation of fibronectin and vimentin levels in ovarian tissues Increased ovarian weight and plasma E2 level	[92]
MSCs	*Mare*	Lack of persistence of MSCs Altered gene expression	Intraovarian	No change in follicle count and serum AMH level No change in oocyte recovery rate, oocyte maturation rate, and blastocyst rate in aged *mares*	[134]
*h*BMSCs	*Human*	Effect of MSC secretome on GCs	Intraovarian (laparoscopy)	50% increased volume in treated ovary 150% increase in E2 Return of menses Marked improvement of menopausal symptoms	[99]
*h*BMSCs	*Human*	Regeneration of oocytes and GCs Reestablished hormone or cytokine profiles	Intraovarian (laparoscopy)	Increased AMH level (0.4 ng/mL to 0.9 ng/mL) 3 oocytes with one Grade A compacting embryo on day 3 2.7 kg female baby with normal karyotype (birth at 38 wks, C/sec)	[104]
*h*BMSCs	*Murine*	Autocrine/paracrine mechanism Local environment (niche)	Intra-arterial catheterization of ovarian artery	Improved ovarian function and AFC in 81.3% of participant 5 pregnancies (two after ET, three by natural conception)	[105]
*h*BMSCs	*Murine*	Paracrine action Altered gene expression Promoting angiogenesis	Intravenous	Increased number of preovulatory follicles, metaphase II oocytes, serum E2, ovarian vascularization, and spontaneous pregnancy Reduced apoptosis	[101]
*h*BMSCs	*Murine*	Restoration of ovarian hormone production Reactivation of follicular genesis	Intraovarian	Increase in total body weight, ovarian weight, and estrogen-responsive organ weight Higher pregnancy numbers, follicle numbers, AMH level, FSH receptor, and inhibin A/B and AMH expression in growing follicles Lower FSH level	[100]
BMSCs	*Murine*	Overexpression of miR-21, which regulates cell apoptosis	Intraovarian	Reduced apoptosis Increased ovarian weight, follicle count, and E2 level Upregulation of miR-21 Downregulation of *PTEN* and *PDCD4*	[107]
BMSCs	*Murine*	Homing to ovarian hilum and medulla Paracrine regulation No direct differentiation	Intravenous	Increased AFC and serum E2 levels	[102]
BMSCs	*Murine*	Protection from germ cell apoptosis and DNA damage	Intravenous	Increased number of primordial follicles	[106]
BMSCs	*Rabbit*	Direct differentiation Paracrine action Angiogenesis	Intravenous	Decreased FSH Increased E2 and VEGF Higher follicle numbers with apparent normal structure of ovarian follicles	[103]
BMSCs	*Murine*	Paracrine mediators secreted by MSC	Intraovarian	Reduced apoptosis of GC Induced upregulation of *Bcl-2* in vivo	[98]
BMSCs	*Murine*	Reactivating host oogenesis Indirect effect on niches	Bone marrow transplantation	Improved long-term fertility (all offspring derived from the recipient germline)	[97]
*h*MenSCs	*Human*	No differentiation Improving oocyte quality by paracrine action	Intraovarian	Increased natural pregnancy and live birth No difference in AMH level, mean AFC, and oocyte number Higher oocyte fertilization rate and embryo number	[115]
MenSCs	*Murine*	Homing to ovarian interstitium No direct differentiation Paracrine action	Intravenous	Reduced apoptosis in GCs and fibrosis of ovarian interstitium Increased number of follicles and E2	[112]
*h*MenSCs	*Murine*	mRNA gene expression pattern in the ovarian cells following stimulation of the host ovarian niche becoming similar to those in *human* ovarian tissue	Intraovarian	Higher AMH, inhibin α/β, FSH receptor, and Ki 67 Increased ovarian weight, plasma E2 level, and the number of normal follicles	[113]
MenSCs	*Murine*	Enhanced support of ovarian SC niche	Intravenous	Increased number of primordial follicles and oocytes Increased pregnancy rate	[114]
*h*MenSCs	*Murine*	Direct differentiation into granulosa cells Restoration of germline SC	Intravenous	Improved cyclicity and fertility	[111]
*h*ADSCs	*Murine*	Improvement of damaged ovarian microenvironment Suppressed activation of the *PI3K*/*Akt*/mTOR axis	Intravenous	Reduced apoptosis of GCs Decreased secretion of FSH Increased number of total follicles, primordial follicles, primary follicles, and mature follicles Increased AMH and E2 Improved ovarian microenvironment	[119]
*h*ADSCs	*Murine*	Regulation of SMAD pathway by exosomes from *h*ADSCs		Increased number of follicles Increased proliferation and decreased apoptosis	[120]
ADSCs	*Murine*	No direct differentiation Altered gene expression regarding follicle formation or ovulation	Intravenous or intraovarian	Higher number of follicles at all stages Decreased apoptosis of GCs	[121]
ADSCs	*Murine*	Inducing angiogenesis Paracrine effect	Intraovarian	Increased secretion of VEGF, IGF-1 and HGF	[72]
UCMSCs	*Human*	Activation of primordial follicles via phosphorylation of FOXO3a and FOXO1	Collagen scaffold transplantation	Elevated E2 concentration Improved follicular development Increased number of antral follicles Successful pregnancy	[123]
*h*UCMSCs	*Murine*	Increased cytokine secretion Paracrine action	Intravenous	Increased number of follicles, AMH and E2 Decreased FSH Increased expression of HGF, VEGF, and IGF-1 protein Improved ovarian structure	[124]
*h*UCMSCs	*Murine*	Anti-apoptotic factors secreted by *h*UMSCs	Intravenous or intraovarian	Improved sex hormone level Restoration of fertility Faster ovarian function recovery in intraovarian group Similar long-term outcome in intravenous and intraovarian groups	[125]
UCMSCs	*Murine*	No direct differentiation Altered RNA expression and signaling pathway	Intravenous	Higher number of follicles Reduced apoptosis of cumulus cells Increased E2 level	[126]
*h*AECs	*Murine*	Paracrine action Upregulation of cytokines involved in apoptosis, angiogenesis, cell cycle and immune response	Intraovarian	Existence of healthy and mature follicles in ovaries treated with *h*AECs or *h*AFC-conditioned medium	[129]
*h*AECs	*Murine*	Homing to ovary Differentiation into GC	Intravenous	Restored ovarian follicle development Increased AMH expression	[76]
AFSCs	*Murine*	AFSCs-derived exosomes delivering miR-146a and miR-10a	Intraovarian	Reduced apoptosis	[131]
*h*AFCs	*Murine*	Homing to ovary Differentiation into GC	Intraovarian	Increased number of follicle-enclosed oocytes at all stages of development Increased AMH expression	[130]
*h*PMSCs	*Murine*	Regulation of Treg cells Production of cytokines	Intravenous	Recovered estrous cycle Increased E2 level Decreased FSH level Decreased apoptosis of GCs Reversed effect of *pZP3* (increased TGF-β and decreased IFN-γ)	[133]
OSCs	*Murine*	Homing Differentiation into early stage oocytes (only when SCs reached the edge of the ovarian cortex)	Intraovarian	Generation of offspring	[135]

POI Premature ovarian insufficiency; *h*ESCs *Human* embryonic stem cells; BMSCs Bone marrow-derived mesenchymal stem cells; hiPSCs *Human* induced pluripotent stem cells; iPSCs Induced pluripotent stem cells; OSE-like cells Ovarian surface epithelium-like cells; ERβ Estrogen receptor beta; E2 Estradiol; MSCs Mesenchymal stem cells; miR-17-3p MicroRNA 17-3p; AMH Anti-Mullerian hormone; *h*BMSCs *Human* bone marrow-derived stem cells; GCs Granulosa cells; C/sec Cesarean section; AFC Antral follicle count; ET Embryo transfer; FSH Follicle-stimulating hormone; miR-21 MicroRNA-21; *PTEN Phosphatase and tensin homolog; PDCD4 Programmed cell death protein 4*; *Bcl-2* B-cell lymphoma 2; Ki 67 Antigen Kiel 67; *h*MenSCs Human menstrual blood mesenchymal stem cells; MenSCs Menstrual blood mesenchymal stem cells; SC Stem cell; *h*ADSCs *Human* adipose tissue-derived stem cells; ADSCs Adipose tissue-derived stem cells; IGF-1 Insulin-like growth factor 1; HGF Hepatocyte growth factor; UCMSCs Umbilical cord-derived mesenchymal stem cells; *h*UCMSCs *Human* umbilical cord-derived mesenchymal stem cells; VEGF Vascular endothelial growth factor; *h*AECs *Human* amniotic epithelial cells; AFSCs Amniotic fluid-derived stem cells; *h*AFCs *Human* amniotic fluid stem cells; *h*PMSCs *Human* placenta-derived mesenchymal stem cells; Treg cells Regulatory T cells; *pZP3 Zona pellucida glycoprotein 3*; OSCs Ovarian stem cells.

## Data Availability

The data presented in this study are available in the article.

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
