# Peer review of "Current Status and Future Prospects of Stem Cell Therapy for Infertile Patients with Premature Ovarian Insufficiency"

_biomolecules, 2024, doi:10.3390/biom14020242_

Round 1
Reviewer 1 Report
Comments and Suggestions for Authors
The author's manuscript, which describes the treatment of today's difficult gynecologic diseases, is very comprehensive and the overall logic is very fluid, but there are also several parts that could be improved.
1. Define the difference between POF,POI(primary ovarian insufficiency),POI(premature ovarian insufficiency).
2. In my opinion it would be appropriate to clarify in the text what the diagnosis of POI is clinically: Although proper diagnostic accuracy in POI is lacking, the GDG recommends the following diagnostic criteria: (i) oligo/amenorrhea for at least 4 months, and (ii) an elevated FSH level .25 IU/l on two occasions .4 weeks apart.
References: European Society for Human Reproduction and Embryology (ESHRE) Guideline Group on POI; Webber L, Davies M, Anderson R, Bartlett J, Braat D, Cartwright B, Cifkova R, de Muinck Keizer-Schrama S, Hogervorst E, Janse F, Liao L, Vlaisavljevic V, Zillikens C, Vermeulen N. ESHRE Guideline: management of women with premature ovarian insufficiency. Hum Reprod. 2016 May;31(5):926-37. doi: 10.1093/humrep/dew027IF: 6.1 Q1 . Epub 2016 Mar 22. PMID: 27008889.
3.In the etiologic analysis, genetic factors can be described in terms of chromosomal abnormalities and genetic abnormalities. Chromosomal abnormalities can be described by autosomal and sex chromosome abnormalities; genetic abnormalities can be described from the causative genes that can cause syndromic POI and the causative genes that can cause non-syndromic POI.
4. In terms of environmental factors contributing to the disease can be cut from lifestyle as well as environmental factors, making it easier to understand the reading.
Very much looking forward to the author's next updates!
Comments on the Quality of English LanguageThe author's manuscript, which describes the treatment of today's difficult gynecologic diseases, is very comprehensive and the overall logic is very fluid, but there are also several parts that could be improved.
1. Define the difference between POF,POI(primary ovarian insufficiency),POI(premature ovarian insufficiency).
2. In my opinion it would be appropriate to clarify in the text what the diagnosis of POI is clinically: Although proper diagnostic accuracy in POI is lacking, the GDG recommends the following diagnostic criteria: (i) oligo/amenorrhea for at least 4 months, and (ii) an elevated FSH level .25 IU/l on two occasions .4 weeks apart.
References: European Society for Human Reproduction and Embryology (ESHRE) Guideline Group on POI; Webber L, Davies M, Anderson R, Bartlett J, Braat D, Cartwright B, Cifkova R, de Muinck Keizer-Schrama S, Hogervorst E, Janse F, Liao L, Vlaisavljevic V, Zillikens C, Vermeulen N. ESHRE Guideline: management of women with premature ovarian insufficiency. Hum Reprod. 2016 May;31(5):926-37. doi: 10.1093/humrep/dew027IF: 6.1 Q1 . Epub 2016 Mar 22. PMID: 27008889.
3.In the etiologic analysis, genetic factors can be described in terms of chromosomal abnormalities and genetic abnormalities. Chromosomal abnormalities can be described by autosomal and sex chromosome abnormalities; genetic abnormalities can be described from the causative genes that can cause syndromic POI and the causative genes that can cause non-syndromic POI.
4. In terms of environmental factors contributing to the disease can be cut from lifestyle as well as environmental factors, making it easier to understand the reading.
Very much looking forward to the author's next updates!
Author Response
The author's manuscript, which describes the treatment of today's difficult gynecologic diseases, is very comprehensive and the overall logic is very fluid, but there are also several parts that could be improved.
REPLY: We are very much thankful to the reviewer for the thorough review. We agree with all the specific comments raised and have revised our paper in light of the useful suggestions. Our responses to the specific comments/suggestions/queries given are provided below.
- Define the difference between POF, POI (primary ovarian insufficiency), POI (premature ovarian insufficiency).
REPLY: Thank you for your comments. The terms premature ovarian insufficiency (POI), primary ovarian insufficiency, premature ovarian failure (POF), and premature menopause can be confusing. Previously, premature ovarian insufficiency (POI) was referred to as premature ovarian failure (POF) or premature menopause. However, because impaired ovarian function can persist for varying lengths of time and the terms "failure" or "menopause" carry negative connotations, the term premature ovarian insufficiency (POI) is now preferred. In our manuscript, we consistently used the nomenclature "premature ovarian insufficiency (POI)" to minimize confusion.
In response to the reviewer's suggestion, we have clarified this terminology in the manuscript's introduction, addressing the need for transparency regarding the interchangeable use of these terms.
- In my opinion it would be appropriate to clarify in the text what the diagnosis of POI is clinically: Although proper diagnostic accuracy in POI is lacking, the GDG recommends the following diagnostic criteria: (i) oligo/amenorrhea for at least 4 months, and (ii) an elevated FSH level 25 IU/l on two occasions, 4 weeks apart.
References: European Society for Human Reproduction and Embryology (ESHRE) Guideline Group on POI; Webber L, Davies M, Anderson R, Bartlett J, Braat D, Cartwright B, Cifkova R, de Muinck Keizer-Schrama S, Hogervorst E, Janse F, Liao L, Vlaisavljevic V, Zillikens C, Vermeulen N. ESHRE Guideline: management of women with premature ovarian insufficiency. Hum Reprod. 2016 May;31(5):926-37. doi: 10.1093/humrep/dew027IF: 6.1 Q1 . Epub 2016 Mar 22. PMID: 27008889.
REPLY: We sincerely appreciate the comprehensive review. In response to the reviewer's feedback, we have carefully revised our manuscript, ensuring the proper inclusion of the reference. The criteria are outlined in the initial paragraph of the introduction (line 39 to line 41). Your valuable insights and comments have been immensely helpful, and we extend our gratitude.
3.In the etiologic analysis, genetic factors can be described in terms of chromosomal abnormalities and genetic abnormalities. Chromosomal abnormalities can be described by autosomal and sex chromosome abnormalities; genetic abnormalities can be described from the causative genes that can cause syndromic POI and the causative genes that can cause non-syndromic POI.
REPLY: Thank you for your valuable comments. In order to enhance the readability and comprehension of the section 2.1 in the etiology, we actively incorporated the reviewer's suggestions and made revisions accordingly. We changed the titles of each section to 2.1.1. X Chromosome Abnormalities, 2.1.2. X Chromosome Genes in POI, and 2.1.3. Autosomal Chromosome Abnormalities, providing more detailed descriptions in each section. Specifically, as pointed out by the reviewer, we elaborated on non-syndromic aspects and provided a more specific explanation of the types of causative genes (lines 89-118).
- In terms of environmental factors contributing to the disease can be cut from lifestyle as well as environmental factors, making it easier to understand the reading.
REPLY: Thank you for the thorough review of our paper. In accordance with the reviewer's suggestion, we have modified the title of section 2.5 to "Lifestyle and Environmental Causes" to better encapsulate the content. Additionally, we have expanded the discussion to include information on other lifestyle factors such as body weight and exercise. Understanding the impact of lifestyle and its association and implications for POI will further the therapeutic management and reducing the burden of the aforementioned health issue.

Reviewer 2 Report
Comments and Suggestions for Authors
the work is well structured and detailed, explodes every single aspect up to the most modern techniques all well summarized in the tables and drawings.
Only note, should explode acronyms ( although many times it has already been done) such as
line 250 HRT does not explode the acronym, line 317, line 288, line 411 (BM). This is just to crown a well done and exhaustive article
Author Response
The work is well structured and detailed, explodes every single aspect up to the most modern techniques all well summarized in the tables and drawings.
Only note, should explode acronyms (although many times it has already been done) such as line 250 HRT does not explode the acronym, line 317, line 288, line 411 (BM). This is just to crown a well done and exhaustive article
REPLY: We are very much thankful to the reviewer for the thorough review. We agree with all the specific comments raised and have revised our paper in light of the useful suggestions. The abbreviations pointed out have all been described in their full terms. Additionally, for other abbreviations, we initially provided the full term before using the abbreviation.
The modified sections are as follows:
Hormone replacement therapy (HRT) at line 275, 279, and 313 (previously at line 250, 256, and 289)
Combined oral contraceptives (OCs) at line 313 and 314 (previously at line 290 and 292)
Stem cells (SCs) at line 342 and 343 (previously at line 317 and 318)
Stem cell therapy (SCT) at line 353 (previously at line 328)
Bone marrow-derived mesenchymal stem cells (BMSCs) at line 437 (previously at line 411)
Induced pluripotent stem cells (iPSCs) at line 448 and 449 (previously at line 423 and 424)
In line with the reviewer's suggestions, we have revised our manuscript to reduce confusion. We appreciate your valuable insights.
